# Cord Blood Metabolite Profiles and Their Association with Autistic Traits in Childhood

**DOI:** 10.3390/metabo13111140

**Published:** 2023-11-09

**Authors:** Christin S. Kaupper, Sophia M. Blaauwendraad, Charlotte A. M. Cecil, Rosa H. Mulder, Romy Gaillard, Romy Goncalves, Ingo Borggraefe, Berthold Koletzko, Vincent W. V. Jaddoe

**Affiliations:** 1The Generation R Study Group, Erasmus MC, University Medical Center Rotterdam, 3000 CA Rotterdam, The Netherlandsr.gaillard@erasmusmc.nl (R.G.);; 2Department of Pediatrics, Sophia’s Children’s Hospital, Erasmus MC, University Medical Center Rotterdam, 3000 CA Rotterdam, The Netherlands; 3Department of Child and Adolescent Psychiatry, Erasmus MC, University Medical Center Rotterdam, 3000 CA Rotterdam, The Netherlands; 4Department of Epidemiology, Erasmus MC, 3000 CA Rotterdam, The Netherlands; 5Molecular Epidemiology, Department of Biomedical Data Sciences, Leiden University Medical Center, 2333 ZC Leiden, The Netherlands; 6Division of Pediatric Neurology, Developmental Medicine and Social Pediatrics, Comprehensive Epilepsy Center for Children and Adolescents, Dr. von Hauner Children’s Hospital, LMU University Hospitals, LMU—Ludwig-Maximilians Universität, 80337 Munich, Germany; 7Division of Metabolic and Nutritional Medicine, Department of Pediatrics, Dr. von Hauner Children’s Hospital, LMU University Hospitals, LMU—Ludwig-Maximilians Universität, 80337 Munich, Germany

**Keywords:** metabolomics, cord blood metabolomics, ASD, autism spectrum disorder, sphingomyelines, non-esterified fatty acids, carnitines

## Abstract

Autism Spectrum Disorder (ASD) is a diverse neurodevelopmental condition. Gene–environmental interactions in early stages of life might alter metabolic pathways, possibly contributing to ASD pathophysiology. Metabolomics may serve as a tool to identify underlying metabolic mechanisms contributing to ASD phenotype and could help to unravel its complex etiology. In a population-based, prospective cohort study among 783 mother–child pairs, cord blood serum concentrations of amino acids, non-esterified fatty acids, phospholipids, and carnitines were obtained using liquid chromatography coupled with tandem mass spectrometry. Autistic traits were measured at the children’s ages of 6 (*n* = 716) and 13 (*n* = 648) years using the parent-reported Social Responsiveness Scale. Lower cord blood concentrations of SM.C.39.2 and NEFA16:1/16:0 were associated with higher autistic traits among 6-year-old children, adjusted for sex and age at outcome. After more stringent adjustment for confounders, no significant associations of cord blood metabolites and autistic traits at ages 6 and 13 were detected. Differences in lipid metabolism (SM and NEFA) might be involved in ASD-related pathways and are worth further investigation.

## 1. Introduction

Autism Spectrum Disorder (ASD) is a diverse neurodevelopmental condition characterized by difficulties in social communication and interaction as well as restrictive and repetitive behavior, interests or activities [1]. Increasing evidence indicates that ASD represents the extreme end of a continuum of autistic traits that are present in the general population [2,3]. The biological mechanisms underlying ASD remain unclear. However, a growing body of evidence suggests that the interplay of genetic and environmental factors may trigger immune and inflammatory responses in early stages of life, which could play a key role in pathophysiological processes leading to ASD [4,5]. These processes might result in metabolic dysregulations that can be observed in cord blood, detectable through metabolomics prior to the manifestation of autistic traits in early childhood. Multiple cross-sectional metabolomics studies among pre-school children with an ASD diagnosis and typically developing children have been conducted and suggested that mitochondrial dysfunction, immune dysregulation, as well as altered amino acid, lipid, and neurotransmitter metabolism could be associated with ASD [6,7,8,9,10,11,12,13,14,15,16,17,18]. Only one previous study assessed the prospective associations of stored dried blood spot metabolites in a small group of 37 newborns that received an ASD diagnosis in childhood and in healthy controls. They did not detect significant differences in metabolite concentrations [19]. The majority of previous studies have been based on small, selected clinical samples using a cross-sectional design, which precluded the possibility of assessing the temporality of associations. In addition, given that social impairments related to ASD exist on a spectrum that extends into the typical range of physiological variations, differences in metabolic composition might also be detectable in fetuses that develop sub-clinical autistic traits in childhood. 

We hypothesize that changes in the blood metabolite composition due to early-life metabolic adaptations may be associated with autistic traits in the general population. We tested this hypothesis on 783 children participating in a population-based prospective cohort study from early fetal life onwards. We used a targeted metabolomics approach for amino acids (AA), non-esterified fatty acids (NEFA), phospholipids (PL), sphingomyelins (SM), and carnitines (Carn) from cord blood. These metabolites play an important role in lipid metabolism, signal transduction, oxidative stress, inflammatory responses, and mitochondrial as well as neurotransmitter metabolism and might therefore be involved in ASD pathophysiology [20,21,22,23]. Autistic traits were measured dimensionally at the ages 6 and 13 with the Social Responsiveness Scale. 

## 2. Methods

### 2.1. Study Design

This study was embedded in the Generation R Study, a prospective population-based cohort from fetal life onwards, designed to identify early environmental and genetic determinants of growth and development. Pregnant women who resided in the Rotterdam area and delivered between April 2002 and January 2006 were eligible for participation. Study approval was obtained by the Medical Ethical Committee of the Erasmus University Medical Center, Rotterdam (MEC 198.782/2001/31). Written informed consent was obtained from all mothers. Umbilical cord blood metabolites were assessed in a subgroup of 921 mother–infant pairs. Of those, 716 and 648 had information on autistic traits at 6 and 13 years, respectively, and 581 had information at both 6 and 13 years (see Flowchart in Appendix A).

### 2.2. Metabolomics Measurements

Umbilical venous cord blood samples were collected by a midwife or an obstetrician (median gestational age at birth: 40.3 weeks; (95% range: 36.9, 42.4)) after birth [24]. Blood samples were transported to the regional laboratory (STAR-MDC), spun, and stored at −80 °C within a maximum of 4 h after collection. For metabolite measurements, the samples were transported on dry ice to the Division of Metabolic and Nutritional Medicine, Dr. von Hauner Children’s Hospital, LMU Munich. There, a targeted metabolomics approach was performed in order to determine the serum concentrations of 193 metabolites (μmol/L), AA, NEFA, PL (including diacyl-lysophosphatidylcholines (PC.aa), acyl-alkyl-lysophosphatidylcholines (PC.ae), acyl-lysophosphatidylcholines (Lyso.PC.a), alkyl-lysophosphatidylcholines (Lyso.PC.e)), SM, and Carn (including free carnitine (Free Carn) and acylcarnitines (Carn.a)). The metabolomics analysis is described in detail elsewhere [24]. In brief, a 1100 high-performance liquid chromatography (HPLC) system (Agilent, Waldbronn, Germany) coupled with a API2000 tandem mass spectrometer (AB Sciex, Darmstadt, Germany) analyzed AA [25]. For the amino acid notation, IUPAC-IUB nomenclature was used [26]. The measurement of PL, NEFA, and Carn was conducted with a 1200 SL HPLC system (Agilent, Waldbronn, Germany) coupled to a 4000QTRAP tandem mass spectrometer from AB Sciex (Darmstadt, Germany) [27,28]. The analytical measurements enable the distinguishment of the total number of double bonds, however, not their position or the distribution of the C-atoms between the fatty acid side chains. For NEFA, PL, and Carn.a, the following notation was used: ‘X:Y’, ‘X’ designates the number of C-atoms of the carbon chains, and ‘Y’ is the total number of double bonds. The ‘a’ is a synonym for acyl chain bound to the backbone via an ester bond (‘acyl-’), whereas ‘e’ stands for an ether bond (‘alkyl’), respectively. For precise measurements, six quality control (QC) samples per batch were consistently measured between study samples. Outliers were excluded, and the coefficients of variation (CV; standard deviation/mean) for each batch (intra-batch) and for all batches (inter-batch) of the QC samples were calculated for each metabolite. We excluded batches with an intra-batch CV higher than 25%, which is in line with previous studies [29,30,31]. Complete metabolite data was excluded for metabolites that had an inter-batch CV higher than 35% or if less than 50% of the batches had passed the QC. The correction of batch effects was carried out by dividing metabolite concentrations by the ratio of intra-batch median and the inter-batch median of the QC samples [31]. If metabolites or participants missed more than 50% of the values, they were excluded [30]. Missing metabolite values of the remaining metabolites and participants were imputed using the Random Forest algorithm (R package missForest) [32]. Further information on the parameters for mass-spectrometry detection and identification of the metabolites can be found in Appendix A.

For the data analysis, individual metabolites were clustered in metabolite groups (AA, NEFA, PC.aa, PC.ae, Lyso.PC.a, Lyso.PC.e, SM, Free Carn, and Carn.a), as well as in metabolite subgroups based on chemical structure and biological relevance (AA: branched-chain amino acids (BCAA), aromatic amino acids (AAA), essential amino acids (EAA), non-essential amino acids (NEAA); NEFA; PC.aa; PC.ae; Lyso.PC.a; Lyso.PC.e; SM: saturated, mono-unsaturated, poly-unsaturated; Carn.a: short-chain, medium-chain, long-chain) [24]. The sum of individual metabolite concentrations per metabolite group was calculated. Moreover, we computed the following ratios: AA ratios, asparagine/aspartic acid (Asn/Asp), and glutamine/glutamic acid (Gln/Glu) as indicators for anaplerosis or replenishing of citric acid cycle metabolites; NEFA.18:1/NEFA.18:0 and NEFA.16:1/NEFA.16:0 ratios as markers of stearoyl-CoA desaturase-1 activity, which is associated with increased fat accumulation and reduced fatty acid oxidation; ΣPC.aa/ΣPC.ae, reflecting oxidative stress, ΣLyso.PC.a/ΣPC.aa as a lipid biomarker of inflammation; (lyso.PC.a.C16:0 + lsyo.PC.a.C18:0)/ΣPC.aa as a proinflammatory biomarker; (lyso.PC.a.C18:1 + lyso.PC.a.C18:2)/ΣPC.aa as an anti-inflammatory biomarker; Carn.a ratios, Carn.a.C16.0/free Carn and Carn.a.C2:0/Carn.a.C16:0 as markers of Carn palmitoyl transferase-1 activity and fatty acid ß-oxidation [33,34,35,36,37,38]. Individual metabolite concentrations, sums, and ratios were square root transformed to obtain normally distributed metabolite concentrations. In order to enable comparison of the effect estimates, individual metabolite concentrations and sums were standardized by calculating standard deviation scores.

### 2.3. Autistic Traits

At the ages of 6 and 13 years, the primary caregiver of the participating children filled out a short version of the Social Responsiveness Scale (SRS). The SRS is a validated instrument for assessing autistic traits on a continuous scale [39]. It represents the caregiver’s observation of the child’s social behavior during the previous 6 months. The SRS is a useful screening tool to identify children who need further ASD-specific diagnostics [40]. The abridged, 18-item version of the SRS has been shown to be an effective tool for generating results consistent with the full version of the SRS [41]. It is designed for children between the ages of 4 and 18. Each item on the SRS is rated from 0 (never true) to 3 (almost always true), covering social, language, and repetitive behavior, with higher scores indicating greater social impairment [39]. Individual item scores for the SRS were summed and weighted by the number of items completed. We used the SRS as a continuous measure of autistic traits in our analyses.

### 2.4. Covariates 

During pregnancy, data on maternal age at enrollment, educational level (higher education yes/no), smoking (non-smoking, smoked until pregnancy was known, continued smoking during pregnancy), alcohol intake (no intake, until pregnancy was known, continued during pregnancy), and folic acid supplement (yes/no) during pregnancy was gathered using questionnaires. A start of folic acid supplementation during the first 10 weeks of pregnancy or preconceptionally was considered positive. Maternal weight and height at enrollment were measured without shoes, and BMI was calculated. Maternal psychopathology during pregnancy was assessed using the Brief Symptom Inventory (BSI). Maternal vitamin D levels were obtained from serum in pregnancy. Vitamin D levels below 50 nmol/L were considered deficient. Information on sex, gestational age at birth, and birth weight according to European growth charts was obtained from medical records. We calculated gestational age and sex-adjusted birth weight standard deviation scores according to growth charts of Niklasson [42]. Data on the children’s age at outcome was obtained when the primary caregiver filled out the SRS.

### 2.5. Statistical Analysis 

First, we conducted a non-response analysis. We compared mother–child pairs of our subsample with all mother–child pairs with metabolomics data available to identify and take into account possible biases caused by loss to follow-up or exclusion criteria (Appendix A). We used a Chi-squared test for categorical variables and a Student’s *t*-test or Mann–Whitney-U test for normally distributed or skewed continuous variables. We explored the correlation between the SRS scores at 6 and 13 years using Spearman’s correlation. Then, we examined the associations of neonatal metabolite ratios, groups, and individual metabolites with the continuous SRS scores at the ages of 6 and 13 using linear regression models. We chose this approach over data reduction or variable selection analysis methods like Principal Component Analysis because our primary goal was to assess the associations of the individual metabolites using regression models on the outcome autistic traits rather than identifying clusters of metabolites that were jointly associated with the outcome. Analyses were only adjusted for child sex and age at outcome in the basic model and additionally adjusted for our preselected confounders in the main model. Confounders were selected based on previous studies and included maternal age, pre-pregnancy BMI, maternal education level, maternal psychopathology, smoking during pregnancy, alcohol consumption during pregnancy, gestational age at birth, and birth weight [13,18,19,43,44]. Several sensitivity analyses were performed. First, we repeated our main analysis, additionally adjusting for vitamin D levels and folic acid supplementation as previous studies suggested associations of folic acid as well as vitamin D deficiency with autistic traits [4]. Second, we explored the influence of individual confounding variables by separately adjusting for each confounder. 

As an exploratory analysis, we ran a linear mixed-effects model to investigate whether changes in autistic traits over time are associated with certain metabolites. Models were fit using the lme4 package [45]. Linear mixed-effects models contain numerous positive features, including modeling of random effects and handling of missing time points. 

To account for multiple testing, we applied Benjamini–Hochberg correction separately for each model, using an overall false discovery rate (FDR) adjusted *p*-value of <0.05. To account for missing values of covariates, we performed multiple imputations using the fully conditional specification method, and pooled results from 25 imputed datasets were reported [46]. The proportion of missing values ranged from 6% (vitamin D deficiency) to 17% (folic acid supplement). All statistical tests were 2-sided. Statistical analyses were conducted using R statistical software version 4.2.2 (R Foundation for Statistical Computing, Vienna, Austria).

## 3. Results

### 3.1. Population Characteristics

The mean maternal age at enrollment was 31.8 (±3.9) years, and the median body mass index was 22.4 (kg/m^2^) (Table 1). Most women were nulliparous (62%), highly educated (66%), did not smoke (79%), and continued drinking alcohol (56%) in pregnancy. A non-response analysis showed that mothers of children included in our analysis were older, had a higher education level, smoked less, drank less alcohol, and used more folic acid supplements during pregnancy as compared to mother–child pairs with metabolomics data available who did not answer the SRS questionnaire (Appendix A). Spearman correlation between SRS scores at age 6 and at age 13 suggested a moderate (intercept of 0.58) correlation. Median concentrations of cord blood metabolite groups, individual metabolites, and metabolite ratios are shown in Appendix A.

### 3.2. Neonatal Cord Blood Metabolomics and Autistic Traits at the Ages of 6 and 13

Higher cord blood SM.a.C39.2 concentrations were significantly associated with a 0.6 lower SRS score at 6 years (95% CI (−0.9, −0.31) per SDS increase in SM.a.39.2, FDR-adjusted *p*-value 0.01) in the basic model (Figure 1A, Table 2), but not in the main model (0.58 lower SRS score (95% CI (−0.84, −0.24) per SDS increase, FDR-adjusted *p*-value 0.08) (Figure 1B, Table 2). The NEFA.16:1/16:0 ratio was inversely associated with SRS scores in the basic model (0.58 lower SRS score (95% CI (−0.83, −0.26)) per SDS increase in NEFA.16:1/16:0 ratio, FDR-adjusted *p*-value: 0.02) (Figure 1A, Table 2) but did not remain significant in the main model after multiple testing correction (0.48 lower SRS score (95% CI (−0.77, −0.19)) per SDS increase in NEFA.16:1/16:0 ratio, FDR-adjusted *p*-value: 0.08) (Table 2). We observed that higher Carn.a.C18.2 concentrations were associated with higher SRS scores at age 6 in the basic (0.48 higher SRS score (95% CI (0.19, 0.77)) per SDS increase in Carn.a.C18.2, FDR-adjusted *p*-value 0.06) and main model (0.51 higher SRS score (95% CI (0.22, 0.8)) per SDS increase in Carn.a.C18.2, FDR-adjusted *p*-value 0.08), although it did not survive multiple testing (Table 2). No significant associations between total metabolite ratios, groups, individual concentrations, and autistic traits at 13 years were observed (Appendix A)). Linear mixed-effects analyses did not yield significant associations between changes in autistic traits over time and metabolites (Appendix A).

### 3.3. Sensitivity Analysis

The additional adjustment for vitamin D deficiency and folic acid supplementation did not influence the associations between cord blood metabolites and SRS scores at age 6 or 13 years (Appendix A). Adjusting for every confounder separately showed that gestational age had the strongest influence on the association between cord-blood metabolites and autistic traits.

## 4. Discussion

In this prospective cohort study, we observed some suggestive evidence for associations between SM, NEFA ratio, and autistic traits at age 6 years, but these associations did not remain significant after multiple testing correction and more rigorous covariate adjustment. No significant associations between cord blood metabolites and autistic traits at the ages of 6 and 13 were observed in our main model. Moreover, no effect modification by vitamin D deficiency or folic acid deficiency was observed. Controlling for gestational age at birth showed the biggest effect modification. Finally, no associations between changes in autistic traits over time and metabolites were detected.

### 4.1. Interpretation of Main Findings

The etiology of ASD is complex and poorly understood, but various metabolic pathway disturbances as a result of gene–environment interactions already occurring during the fetal period are believed to contribute to this heterogeneous condition. Cord blood metabolomics is able to capture the metabolic phenotype as a result of the interplay between genetics and environment. Moreover, it presents combined information on both maternal metabolism transferred via the placenta and fetal metabolism [30]. Thus, it may contribute to elucidating the complex etiology of ASD. A review from 2020 that included 10 case–control studies reported repeatedly identified differences in amino acid concentrations (tryptophan, BCAA), pathways of oxidative damage (taurine level elevations), abnormalities in the nitric oxide pathway, and disturbed lipid metabolism (decreased levels of lysolipids, free- and short-/long-chain Carn.a, docosahexaenoic acid, increased levels of sphingosine-1-phosphate), as well as disruptions of the Krebs cycle, oxidative phosphorylation, and respiratory chain among pre-school children with an ASD diagnosis compared to non-ASD-diagnosed controls [9]. Moreover, some studies explored the associations between metabolomics and autistic traits prospectively. An untargeted metabolomics case–control study that analyzed the blood of 30 mothers whose children received an ASD diagnosis and of 30 controls detected, amongst others, differences in histidylglutamate, cinnamoglycine, proline, and adrenoylcarnitine concentrations [43]. Another case–control study analyzed stored mid-pregnancy blood of 52 women whose children later received an ASD diagnosis and of 62 controls [13]. They reported differences in several glycosphingolipid, n-glycan, and pyrimidine metabolism pathways. Together, those studies suggest that alterations in the metabolic composition of the blood might be present in children diagnosed with ASD from early life onwards. However, the metabolites that showed differential abundance varied across the studies. Possible explanations could be the lack of uniformity in the collection and storage of serum samples as well as different methods in obtaining the metabolite concentrations (Nuclear Magnetic Resonance (NMR) vs. liquid chromatography–mass spectrometry/mass spectrometry (LC-MS/MS)). Moreover, some studies, such as ours, used a targeted metabolomics approach with a predefined set of metabolites, whereas other studies used an untargeted approach. Finally, varying ethnicity, age of the study population, and different dietary habits could have influenced the results and, therefore, contributed to its heterogeneity. 

We are the first study to examine the prospective associations of cord blood metabolites and autistic traits in childhood dimensionally. No associations between metabolites at birth and autistic traits at ages 6 and 13 were observed in our main model. Autism is a heterogeneous condition with a complex etiology that is only partly elucidated, and not all pathophysiological mechanisms may be detectable through changes in the metabolic composition of the blood. Furthermore, the neurodevelopment of the brain extends after birth, and processes that contributed to an autistic condition in childhood might not have been present at birth. Moreover, the blood metabolome is influenced by contemporary environmental factors such as diet and stress level and may therefore derive stronger associations in cross-sectional settings. A prior study conducted within the Generation R Study revealed that correlations between metabolite measurements at various time points (early pregnancy, cord blood, and at 10 years) were low, ranging between r = −0.10 and r = 0.35 [24]. Observed associations in our study attenuated from 6 to 13 years, and the linear mixed-effects model did not yield associations on changes in autistic traits over time with metabolites. As the SRS is parent-reported, puberty could have caused difficulties in assessing children’s behavior correctly compared to age 6. Further, after the age of 6, other environmental influences could have caused the presence of autistic traits at age 13 and thus weakened its association with metabolites. Our results are in line with a Danish case–control study that investigated the association of dried blood spot metabolites obtained 6 days after the birth of 37 ASD cases and 37 healthy controls [19]. They did not detect significant differences in metabolite concentrations among children with an ASD diagnosis as compared to healthy controls. Corroborating our results, the Danish study also reported that gestational age at birth has a strong confounding effect on the association between metabolomics and ASD. A study that analyzed weekly blood samples from 30 pregnant women was able to predict the gestational age of fetuses from blood metabolite measurements [47]. Therefore, future studies involving cord blood metabolomics should consider gestational age as a confounding variable. 

However, our results provide suggestive evidence for an inverse association between the concentration of SM.a.C39.2 at birth and autistic traits at age 6 years. This association was observed after multiple testing correction in our basic model but did not survive correction after adjusting for a wider set of confounders, even though effect sizes were comparable. While this finding should be interpreted with caution, it is noteworthy that SM has previously been implicated in neurodevelopmental outcomes and ASD [17,48,49]. SM is highly abundant in the brain as part of the myelin isolating oligodendrocytes [50]. It comprises cellular membranes and is involved in proliferation, migration, inflammation, and cell survival [51]. A small randomized controlled trial among 24 very-low-birth-weight infants reported improvements in neurodevelopmental tests after feeding them SM-fortified milk [52]. Moreover, an intervention study of 15 ASD-diagnosed children reported correlations between increased urinary levels of SM and improvement of autistic traits after treatment with sulforaphane [53]. However, SM.a.C39.2 is an unusual SM in relation to brain myelination, as most SM-forming myelin has between 18 and 24 C-atoms [54]. Moreover, SM.a.C39.2 contributes only about 0.2% to the total SM in our samples. Therefore, a potential deviation of early-life sphingomyelin formation during brain development could be a target for further exploration in relation to the risk of ASD.

Numerous studies have reported differences in fatty acid concentration, Carn.a, and PL, which are linked to mitochondrial dysfunction, in ASD [43,49,55,56,57]. Carn.a transport acyl groups into the mitochondrial matrix for ß-oxidation. Thus, it is crucial crucial for mitochondrial energy production [20]. Previous studies suggest that the accumulation of Carn.a could be a result of incomplete oxidation of fatty acids due to mitochondrial dysfunction [58]. In our models, although non-significant, a trend between elevated Carn.a.C.18.2 concentrations and higher autistic traits at age 6 was observed in the main model. 

This is the first autism-related study to report a decreased NEFA16:1/16:0 ratio associated with higher severity of autistic traits, albeit not significant after multiple testing correction in the main model. The ratio represents a proxy of the stearoyl-CoA desaturase-1 (SCD-1) activity. This enzyme breaks down saturated fatty acids, and decreased levels of SCD-1 have been previously linked to inflammation in, e.g., macrophages and endothelial cells [33,59]. Inflammatory processes have been previously linked to ASD pathophysiology [60]. Moreover, SCD-1 functions to form mono-unsaturated fatty acids, of which relatively large amounts are deposited in brain lipids during early brain growth [54]. Therefore, it is required for normal brain development. Reduced levels of NEFA16:1/16:0 ratio could reflect neurodevelopmental abnormalities occurring during pregnancy, possibly leading to a higher abundance of autistic traits in childhood.

In our population-based study, we did not identify certain metabolites at birth associated with autistic traits in childhood. Although we found some suggestive evidence that decreased SM, NEFA16:1/16:0 concentrations and increased Carn.a concentrations at birth might be predictive of more severe autistic traits in childhood, associations attenuated after adjusting for a wider set of confounders and disappeared at the age of 13. Thus, we conclude that the measured metabolites within our study are not suitable for the prediction of autistic traits in childhood. For future studies, it might be beneficial to explore associations with a wider set of metabolites or even consider an untargeted approach. Moreover, repeated measurements on both metabolomics and autistic traits would enable us to study their association over time.

### 4.2. Methodological Considerations

Among the strengths of this study is the dimensional measure of autistic traits with the SRS score. Although our dimensional approach does not provide a clinical diagnosis, it allows us to account for children with fewer symptoms and model the full range of symptoms in the general population. It also reduces the risk of outcome misclassification and increases power. Moreover, obtaining measurements of SRS scores at two time points is another strength. It allowed us to study the association of longitudinal trajectories of autistic traits with metabolites. Finally, the availability of sociodemographic and lifestyle information of our participants allowed us to control for a broad range of confounders.

Some limitations in the current study need to be considered. Unlike previous studies that used a case–control design, our study was embedded in a population-based cohort. Although the percentage of participants above the recommended SRS cutoff value (0.9%) corresponds to the ASD prevalence (0.6–2.2%), it is noteworthy that our participant’s autistic traits primarily fall within a subclinical range [4]. Consequently, the identified associations may not be as distinct as in case–control designs. Metabolomics data of the Generation R cohort was available for a subgroup of individuals of Dutch ethnicity, with highly educated, normal-weight mothers who were less likely to drink or smoke during pregnancy. This circumstance might have resulted in a selection bias, causing less severe autistic traits among children. We used a targeted metabolomics approach and analyzed 193 metabolites. By limiting our analyses to a predefined set of metabolites, we might have missed associations of other metabolites with autistic traits. Finally, despite adjusting for many potential confounders, due to the observational nature of our study, we cannot completely rule out the effect of residual confounding.

## 5. Conclusions

We did not find significant associations between cord blood metabolite profiles and autistic traits at the ages of 6 and 13 in our main model. Our results did provide suggestive evidence for associations between SM and NEFA16:1/16:0 abundance at birth and subclinical autistic traits at age 6. These findings should be regarded as hypothesis-generating and require further investigations. 

## Figures and Tables

**Figure 1 metabolites-13-01140-f001:**
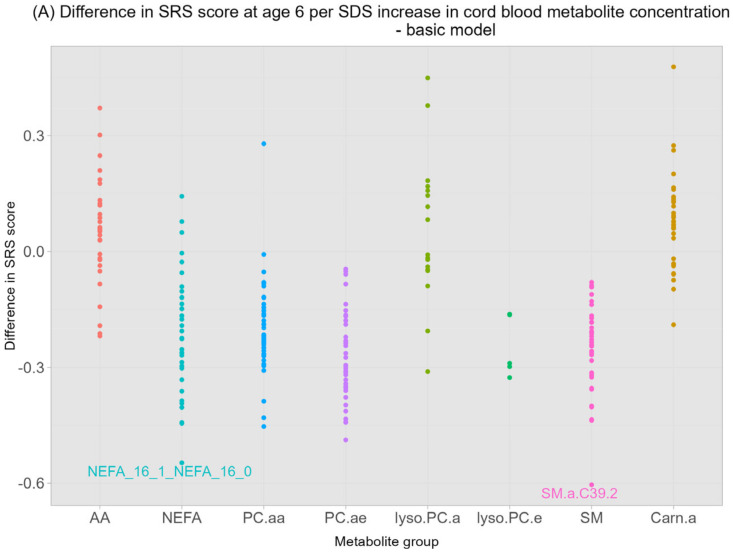
Associations of individual cord-blood metabolites with SRS scores at the age of 6 years. (**A**) basic model, (**B**) main model. Values represent the estimated change in the SRS score associated per SDS increase of a single cord blood metabolite (µmol/L) from linear regression models. (**A**) is adjusted for sex and age at outcome; (**B**) is adjusted for sex, age at outcome, maternal BMI, maternal psychopathologies, education level, smoking during pregnancy, alcohol intake during pregnancy, gestational age at birth, and birthweight. Labeled values represent significant associations (FDR-adjusted *p*-values < 0.05). Corresponding numerical values are shown in Appendix A. AA amino acids, NEFA non-esterified fatty acids, PC.aa diacyl-phosphatidylcholines, PC.ae acyl-alkyl-phosphatidylcholines, lyso.PC.a. acyl-lysophosphatidylcholines, lyso.PC.e alkyl-lysophosphatidylcholines, Carn.a acylcarnitines, SM sphingomyelines. Each color represents a different metabolite group, e.g., red represents “AA”.

**Table 1 metabolites-13-01140-t001:** General characteristics of the study population.

Characteristics	Total Sample *n* = 783
Maternal characteristics	
Age at enrolment mean (±SD), years	31.8 (3.9)
Education level, high, *n* (%)	510 (65.1%)
Pre-pregnancy body mass index, median (95% range), kg/m^2^	22.4 (18.5, 34.0)
Smoking, *n* (%)	
Never smoked during pregnancy	555 (79.1%)
Smoked until pregnancy was known	63 (9.0%)
Continued smoking during pregnancy	84 (11.9%)
Alcohol use, *n* (%)	
No alcohol consumption during pregnancy	207 (29.6%)
Alcohol consumption until pregnancy was known	103 (14.7%)
Alcohol consumption continued during pregnancy	389 (55.7%)
Psychopathology, median (95% range)	0.1 (0, 1)
Folic acid supplements use, yes, *n* (%) *	600 (92.9%)
Vitamin D deficiency, yes, *n* (%) **	221 (30.2%)
Fetal characteristics	
Fetal sex, female, *n* (%)	372 (47.5%)
Gestational age at birth in weeks, median (95% range)	40.3 (36.9, 42.4)
Birthweight in grams, mean (±SD)	3541 (494.0)
Birthweight < 2500 g, *n* (%)	16 (2.0%)
Birthweight 2500 to 4500 g, *n* (%)	744 (95.0%)
Birthweight > 4500 g, *n* (%)	23 (3.0%)
Child characteristics	
6 years visit	
Age at visit in years, median (95% range)	5.9 (5.7, 6.8)
SRS autistic traits score, median (95% range)	3.0 (0, 12)
13 years visit	
Age at visit in years, median (95% range)	13.5 (13, 14.4)
SRS autistic traits score, median (95% range)	4.0 (0, 15)

SRS: Social Responsiveness Scale. Values presented as mean (±standard deviation (SD), median (interquartile range (95% range)), or number of participants (valid%). Psychopathology: GSI score; Number of missing per covariate: smoking, *n* = 81 (10%); alcohol use, *n* = 84 (11%); psychopathology, *n* = 51 (7%); education, *n* = 7 (1%); folic acid supplement *n* = 137 (17%), vitamin D, *n* = 50 (6%). * Start of folic acid supplementation preconception or in first 10 weeks of pregnancy = “yes”. ** Vitamin D deficiency: <50 nmol/L.

**Table 2 metabolites-13-01140-t002:** Association of cord-blood metabolites and SRS scores at age 6 and 13.

	SRS Age 6				SRS Age 13			
	Basic Model		Main Model		Basic Model		Main Model	
	Beta	*p*-Value *	Beta	*p*-Value *	Beta	*p*-Value *	Beta	*p*-Value *
Sphingomyelines SM.a.C.39.2	−0.60	0.01	−0.54	0.08	−0.05	0.88	0.02	0.99
Carnitines Carn.a.C.18.2	0.48	0.06	0.51	0.08	0.28	0.51	0.31	0.69
NEFA Ratio NEFA 16:1/16:0	−0.55	0.02	−0.48	0.08	−0.38	0.40	−0.32	0.69

* FDR-corrected *p*-value. Basic model: adjusted for gender and age at outcome; main model: adjusted for gender, age at outcome, maternal BMI, maternal psychopathologies, education level, smoking during pregnancy, alcohol intake during pregnancy, gestational age at birth, and birthweight.

## Data Availability

The data presented in this study are available on request from the corresponding author. The data are not publicly available due to privacy and ethical restrictions.

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
