# Peer review of "Cord Blood Metabolite Profiles and Their Association with Autistic Traits in Childhood"

_metabolites, 2023, doi:10.3390/metabo13111140_

Round 1

Reviewer 1 Report

Comments and Suggestions for Authors

Manuscript: metabolites-2663432

I want to thank the Multidisciplinary Digital Publishing Institute and Tanongsak Laowanitwattana, Section Managing Editor, for the opportunity to review this article. 

I would also like to thank the authors of this article for their effort and pursuit of understanding changes in the blood metabolite composition due to early-life metabolic adaptations that may be associated with autistic traits in the general population. 

The authors present results of a prospective population-based cohort of 921 mother-infant pairs from fetal life onwards to identify early environmental and genetic determinants that could be related to autistic traits. 

General: 

This data is interesting, and much needs to be reported. 

I congratulate the authors for this enormous amount of effort and work. 

Abstract: 

line 32-34: Eliminate from “our ……. to adjustment”

line 36: Eliminate “when conducting …………….. studies”

Title: 

I recommend leaving the title: “Cord Blood Metabolite Profile and its Association to Autistic Traits in Childhood.” 

Results:

No comments

Discussion: 

The discussion is concise, sound, and well-written. 

However, the authors should correct the minor spelling errors. 

Line 265: I recommend changing the term “did not survive” to “but these associations did not remain significant with a more rigorous covariate adjustment after multiple testing corrections.” 

Line 355: please change “even though did not survive” to “albeit not significant with multiple testing corrections with the main model.”

Conclusion: 

Please erase lines 401 to 404 and from lines 406 to 408. 

The conclusion should be left as follows: 

“We did not find significant associations between cord blood metabolite profiles and autistic traits at 6 and 13 years old in our main model. Our results did provide suggestive evidence for associations between SM and NEFA16:1/16:0 abundance at birth and subclinical autistic traits at age 6. These findings should be regarded as hypothesis generating and require further investigations.“

Comments on the Quality of English Language

Good English, with some minor spelling errors. 

Author Response

I want to thank the Multidisciplinary Digital Publishing Institute and Tanongsak Laowanitwattana, Section Managing Editor, for the opportunity to review this article.  

I would also like to thank the authors of this article for their effort and pursuit of understanding changes in the blood metabolite composition due to early-life metabolic adaptations that may be associated with autistic traits in the general population. The authors present results of a prospective population-based cohort of 921 mother-infant pairs from fetal life onwards to identify early environmental and genetic determinants that could be related to autistic traits. This data is interesting, and much needs to be reported. I congratulate the authors for this enormous amount of effort and work. 

Response: We thank the reviewer for the comments. 

Comment 1: Abstract 

Line 32-34: Eliminate from “our......to adjustment” 

Line 36: Eliminate “when conducting …........ studies” 

Response: We made the following textual changes to the Abstract: 

‘Differences in lipid metabolism (SM and NEFA) might be involved in ASD related pathways and are worth further investigations.’ (Page 1, lines 32-33) 

Comment 2: Title 

I recommend leaving the title: “Cord Blood Metabolite Profile and its Association to Autistic Traits in Childhood.” 

Response: We changed the title accordingly to: ‘Cord Blood Metabolite Profiles and its Association with Autistic Traits in Childhood’ (Page 1, lines 2-3) 

Comment 3: Discussion 

The discussion is concise, sound, and well-written. However, the authors should correct the minor spelling errors. Line 265: I recommend changing the term “did not survive” to “but these associations did not remain significant with a more rigorous covariate adjustment after multiple testing corrections.” 

Line 355: please change “even though did not survive” to “albeit not significant with multiple testing corrections with the main model.” 

Response: We have made the following textual changes in the Discussion section: ‘In this prospective cohort study, we observed some suggestive evidence for associations between SM, NEFA ratio and autistic traits at age 6 years, but these associations did not remain significant after multiple testing correction and more rigorous covariate adjustment.’ (Page 7, lines 278-281) 

‘We are the first autism-related study to report decreased NEFA16:1/16:0 ratio associated with higher severity of autistic traits albeit not significant after multiple testing correction in the main model.’ (Page 9, lines 372-374) 

Comment 4: Conclusion 

Please erase lines 401 to 404 and from lines 406 to 408.  

The conclusion should be left as follows:  

“We did not find significant associations between cord blood metabolite profiles and autistic traits at 6 and 13 years old in our main model. Our results did provide suggestive evidence for associations between SM and NEFA16:1/16:0 abundance at birth and subclinical autistic traits at age 6. These findings should be regarded as hypothesis generating and require further investigations.“ 

Response: We have made the following textual adjustments to the Conclusion section: ‘We did not find significant associations between cord blood metabolite profiles and autistic traits at the age of 6 and 13 in our main model. Our results did provide suggestive evidence for associations between SM and NEFA16:1/16:0 abundance at birth and subclinical autistic traits at age 6. These findings should be regarded as hypothesis generating and require further investigations.’ (Page 10, lines 421-425)

Reviewer 2 Report

Comments and Suggestions for Authors

This manuscript, “Associations of Cord Blood Metabolite Profiles and Autistic Traits in Childhood”, aimed to investigate the association between cord blood metabolome and childhood ASD using targeted metabolomics. Overall, the data analysis was appropriate, and the findings provided useful insights. The following comments or suggestions, if can be addressed, would strengthen this manuscript.

  1. It seems two main figures (Figure 1 and Figure 2) are missing.
  2. The quality and readability of the figure should be improved.
  3. Multivariate analyses, either unsupervised or supervised (e.g., PCA, PLS-DA, etc.), may pick otherwise unobservable ASD-related metabolites.
  4. It would be helpful if the authors could include basic annotation information about the metabolites (e.g., m/z, retention time, etc.)

Author Response

This manuscript, “Associations of Cord Blood Metabolite Profiles and Autistic Traits in Childhood”, aimed to investigate the association between cord blood metabolome and childhood ASD using targeted metabolomics. Overall, the data analysis was appropriate, and the findings provided useful insights. The following comments or suggestions, if can be addressed, would strengthen this manuscript. 

Response: We thank the reviewer for the comments. 

Comment 1: It seems two main figures (Figure 1 and Figure 2) are missing. 

Response: We adjusted the numbering of the figures and changed the subheading of figure 1: Figure 1. (Page 6, line 238; Page 7, line 239) 

Comment 2: The quality and readability of the figure should be improved. 

Response: We adjusted the size of the Metabolite names, the caption of the x and y axis and the color of the background of the figure to improve quality and readability of the figure. 

Comment 3: 

Multivariate analyses, either unsupervised or supverised (e.g., PCA, PLS-DA, etc.) may pick otherwise unobservable ASD-related metabolites. 

Response: We did not apply PCA, as this is an unsupervised method to identify clusters of metabolites within a dataset, independent of the outcome. Our aim was to identify individual metabolites (rather than clusters), associated with autistic traits in childhood. We further clarified this in the Methods section: ‘We chose this approach over data reduction or variable selection analysis methods like Principal Component Analysis because our primary goal was to assess the associations of the individual metabolites using regression models on the outcome autistic traits, rather than identifying clusters of metabolites that were jointly associated with the outcome.’ (Page 4, lines 178-182) 

Comment 4: 

It would be helpful if the authors could include basic annotation information about the metabolites (e.g., m/z, retention time, etc.). 

Response: We provide further information on the parameters for mass-spectrometry detection and identification of the metabolites in the Supplementary material (Supplemental Text T1). Moreover, we added a Supplemental Table (S10) where more detailed information is provided. We referred to it in the Methods section: ‘Further information on the parameters for mass-spectrometry detection and identification of the metabolites can be found in Supplemental Text T1 and Supplemental Table S10.’ (Page 3, lines 116 – 118).

Reviewer 3 Report

Comments and Suggestions for Authors

The authors present a prospective cohort study on cord blood metabolomics on autism spectrum disorders even though they observed negative results/associations. This is a robust, large-scale, population-based study with good arguments in the discussion. The statistics are proper and the employment of the linear mixed-effect model provides suggestive evidence of the prognosis of ASD. 

Some minor points: there are many acronyms in the text that are not spelled out like BCAA, NMR, LC-MS/MS and so on.

Author Response

The authors present a prospective cohort study on cord blood metabolomics on autism spectrum disorders even though they observed negative results/associations. This is a robust, large-scale, population-based study with good arguments in the discussion. The statistics are proper and the employment of the linear mixed-effect model provides suggestive evidence of the prognosis of ASD. 

Response: We thank the reviewer for the comments. 

Comment 1: 

Some minor points: there are many acronyms in the text that are not spelled out like BCAA, NM, LC-MS/MS and so on. 

Response: We added the complete spelling of NMR and LC-MS/MS in the main text as well in the abbreviation table. We made the following textual changes in the Discussion section: ‘Possible explanations could be the lack of uniformity in collection and storage of serum samples as well as different methods in obtaining the metabolite concentrations (Nuclear Magnetic Resonance (NMR) vs. Liquid chromatography – mass spectrometry/mass spectrometry (LC-MS/MS)).’ (Page 8, lines 301-303).  

As for BCAA, the first time the abbreviation was introduced, we spelled the whole term out.: ‘For the data analysis, individual metabolites were clustered in metabolite groups (AA, NEFA, PC.aa, PC.ae, Lyso.PC.a, Lyso.PC.e, SM, Free Carn and Carn.a), as well as in metabolite subgroups based on chemical structure and biological relevance (AA: branched-chain amino acids (BCAA9, aromatic amino acids (AAA), essential amino acids (EAA),non-essential amino acids (NEAA); NEFA; PC.aa; PC.ae; Lyso.PC.a; Lyso.PC.e; SM: saturated, mono-unsaturated, poly unsaturated; Carn.a: short-chain, medium-chain, long-chain). (Page 3, lines 119-125)